# Prediction of Efficacy for Atezolizumab/Bevacizumab in Unresectable Hepatocellular Carcinoma with Hepatobiliary-Phase Gadolinium Ethoxybenzyl-Diethylenetriaminepentaacetic Acid MRI

**DOI:** 10.3390/cancers16122275

**Published:** 2024-06-19

**Authors:** Hideki Kunichika, Kiyoyuki Minamiguchi, Tetsuya Tachiiri, Kozo Shimizu, Ryosuke Taiji, Aya Yamada, Ryota Nakano, Mariko Irizato, Satoshi Yamauchi, Aki Marugami, Nagaaki Marugami, Hayato Kishida, Hiroyuki Nakagawa, Megumi Takewa, Ken Kageyama, Akira Yamamoto, Eisuke Ueshima, Keitaro Sofue, Ryuichi Kita, Hiroyuki Kurakami, Toshihiro Tanaka

**Affiliations:** 1Department of Diagnostic and Interventional Radiology, Nara Medical University, Kashihara 634-8522, Japan; k102972@naramed-u.ac.jp (H.K.); totanaka@naramed-u.ac.jp (T.T.); 2Central Division of Radiology, Nara Medical University, Kashihara 634-8522, Japan; 3Department of Radiology, Nara Prefecture General Medical Center, Nara 630-8054, Japan; 4Department of Radiology, Nara Prefecture Seiwa Medical Center, Sango 636-0802, Japan; 5Department of Diagnostic and Interventional Radiology, Osaka Metropolitan University, Osaka 545-0051, Japan; 6Department of Radiology and Center for Endovascular Therapy, Kobe University, Kobe 650-0017, Japan; 7Department of Gastroenterology and Hepatology, Osaka Red Cross Hospital, Osaka 543-8555, Japan; 8Institute for Clinical and Translational Science, Nara Medical University, Kashihara 634-8522, Japan

**Keywords:** hepatocellular carcinoma, atezolizumab plus bevacizumab, magnetic resonance imaging, hepatobiliary phase, coefficient of variation

## Abstract

**Simple Summary:**

Atezolizumab/bevacizumab (AB) therapy is currently one of the first-line drugs for patients with unresectable hepatocellular carcinoma (u-HCC). Predicting early progressive disease before AB therapy may be beneficial for selecting alternative treatments for patients with u-HCC. Coefficient of variation (CV), as one of the quantitative indicators of signal heterogeneity, in the hepatobiliary phase of Gd-EOB-DTPA-MRI was an independent predictive factor for tumor progression (*p* = 0.043). Patients with a high CV tended to have shorter PFS than those with a low CV (3.5 vs. 6.7 months, *p* = 0.071). Quantitative analysis using CV may be useful for predicting tumor progression for AB therapy, contributing to the individualization of therapeutic strategies for patients with u-HCC.

**Abstract:**

Background: This study aimed to examine whether the coefficient of variation (CV) in the hepatobiliary-phase (HBP) of Gd-EOB-DTPA-MRI could be an independent predictive factor for tumor progression. Methods: Patients who underwent Gd-EOB-DTPA-MRI before Atezolizumab/bevacizumab therapy at six affiliated institutions between 2018 and 2022 were included. CV for each patient was calculated as the mean value for up to five tumors larger than 10 mm, and CV of the whole tumor was calculated using LIFEx software. The tumor response was evaluated within 6–10 weeks. The primary endpoint was to investigate the predictive factors, including CV, related to tumor progression using logistic regression analysis. The secondary endpoints were tumor response rate and progression-free survival (PFS) based on CV. Results: Of the 46 enrolled patients, 13 (28.3%) underwent early progressive disease. Multivariate analysis revealed that a high CV (≥0.22) was an independent predictive factor for tumor progression (*p* = 0.043). Patients with a high CV had significantly frequent PD than those with a low CV (43.5 vs. 13.0%, *p* = 0.047). Patients with a high CV tended to have shorter PFS than those with a low CV (3.5 vs. 6.7 months, *p* = 0.071). Conclusion: Quantitative analysis using CV in the HBP of Gd-EOB-DTPA-MRI may be useful for predicting tumor progression for atezolizumab/bevacizumab therapy.

## 1. Introduction

The advent of immunotherapy has heralded a new era in the treatment of unresectable hepatocellular carcinoma (u-HCC). Atezolizumab plus bevacizumab (AB) therapy, a combination of immunotherapy and anti-angiogenic therapy, is a promising first-line therapy for u-HCC, according to the latest Barcelona Clinic Liver Cancer (BCLC) guidelines—a result of the IMbrave 150 clinical trial [1]. However, Ducreux et al. reported a treatment response by Response Evaluation Criteria In Solid Tumors version 1.1 (RECIST 1.1) is associated with overall survival (OS) and is an independent predictor of OS. Their findings showed that OS was extremely short in those with progressive disease (PD) (6.8 months) compared with those with partial response (PR) or complete response (CR) (26.2 months) and those with stable disease (SD) (17.1 months), per RECIST 1.1 [2]. Therefore, predicting PD per RECIST 1.1 before treatment is important because other treatment options may be candidates for u-HCC.

The hepatobiliary phase (HBP) of gadolinium ethoxybenzyl-diethylenetriaminepentaacetic acid-enhanced magnetic resonance imaging (Gd-EOB-DTPA-MRI) is expected to contribute to the prediction of immunotherapy treatment efficacy [3,4,5]. HCC with high signal intensity (SI) in HBP is reported to be associated with β-catenin–activated HCC, which is resistant to immunotherapy and has a poor clinical outcome in monotherapy and combined immunotherapy and anti-angiogenic therapy [3,6,7,8]. Although the relative enhancement ratio (RER) is often used to quantify the level of high SI in HBP, it is strongly influenced by the background of the liver parenchyma. Consequently, RER may be unreliable in patients with advanced liver cirrhosis. Therefore, it is necessary to develop indicators that are more accurate than the RER to predict the treatment efficacy of immunotherapy.

HCCs with heterogeneous SI on HBP have also been reported to have poor clinical outcomes for AB therapy. The HBP of EOB-MRI may be useful for predicting the therapeutic effect. Tumor signal heterogeneity must be quantitatively assessed; however, only visual assessments have been performed [7].

Coefficient of variation (CV), a quantitative indicator of signal heterogeneity, has a wide variety of applications in the field of oncology, ranging from diagnosis to prediction of treatment response and prognosis [9,10,11,12,13,14,15]. Emori et al. reported CV, representing tumor signal heterogeneity, is an independent predictor of overall survival (OS) in malignant peripheral nerve sheath tumors [16]. Minamiguchi et al. reported that CV in the HBP is useful for predicting prognosis following trans-arterial chemoembolization (TACE) in intermediate-stage HCC [14]. We hypothesized that CV, which is a quantitative and non-visual indicator of heterogeneity, could predict the initial treatment outcomes as one of the prognostic factors to the AB therapy in u-HCC. This study aimed to examine whether the CV of the whole tumor is an independent predictive factor for tumor progression after AB therapy for u-HCC.

## 2. Materials and Methods

### 2.1. Patients and Study Design

The study protocol was approved by the Institutional Ethics Committee of Nara Medical University (approval no. 3331). After receiving official approval, this study was conducted as a retrospective analysis of database records based on the Guidelines for Clinical Research issued by the Ministry of Health and Welfare, Japan.

We retrospectively reviewed patients with u-HCC from six affiliated institutions. The inclusion criteria were patients who were treated with AB therapy between September 2018 and January 2022, with Gd-EOB-DTPA-MRI obtained within 6 weeks before treatment, those whose treatment response was assessed by Gd-EOB-DTPA-MRI or contrast-enhanced computed tomography (CECT) within 6–10 weeks after the initiation of the AB therapy, those with Child–Pugh classification A or B.

The diagnosis of HCC is established based on CT and/or MRI findings. To measure CV, we selected tumors larger than 10 mm and those that had not been treated with topical treatment such as trans-arterial chemoembolization or radiofrequency ablation.

Clinical characteristics and patient information, including sex, age, performance status, etiology of liver disease, Child–Pugh class, albumin–bilirubin (ALBI) score, alpha-fetoprotein (AFP) level, and previous treatment history for HCC, were obtained from medical records. Patients were classified using not only the BCLC guidelines but also up-to −7 and −11 criteria [17,18]. Patients were followed up from the initiation of AB therapy to the date of death or last survival date. Progression-free survival (PFS) was defined as the length of time during and after AB therapy in patients who lived with the disease but whose status did not worsen. Overall survival (OS) was defined as the length of time from the start date of AB therapy until the date of death.

AB therapy was administered according to pharmaceutical recommendations. Patients received 1200 mg of atezolizumab and 15 mg/kg intravenously every 3 weeks as a standard dose.

Among the clinical and radiological findings, the risk factors for tumor progression, including CV and RER, were examined using multivariate analysis. In addition, we compared the response rates and PFS between the high- and low-CV groups.

### 2.2. Pretreatment Gd-EOB-DTPA-MRI

MRI examinations were performed using the systems listed in Appendix A. Post-contrast fat-suppressed T1-weighted images were acquired using the following protocol: Gd-EOB-DTPA (Primovist; Bayer Healthcare, Berlin, Germany) was intravenously injected at a dose of 0.1 mL/kg (0.025 mmol/kg).

### 2.3. Imaging Analysis: CV

An outline of the imaging analysis is shown in Figure 1. Image normalizations were performed to obtain comparable features from different HBP images. In the preprocessing step, the N4 bias field correction algorithm was applied to correct the inhomogeneity of images using the 3D Slicer software (version 5.6.1), a free, open source software package; https://www.slicer.org, accessed on 1 September 2023.

Subsequently, two board-certified radiologists (N.M. and T.T., with 28 and 9 years of experience, respectively) specializing in abdominal imaging, who were blinded to the clinical information, manually drew the volumes of interest (VOIs) to include the whole tumor volume on multiple slices using LIFEx software (version 7.3.0; https://www.lifexsoft.org), accessed on 1 September 2023 [19]. All images were simultaneously segmented and verified by consensus. We selected up to five hyper-vascularized tumors >10 mm and 64 voxels per patient. We excluded tumors that were treated with therapeutic interventions such as TACE and those with necrosis or hemorrhage. The settings used to calculate the CV of the entire images were as follows: spatial resampling using 2.0 × 2.0 × 2.0 mm^3^ for spacing, intensity discretization using 64 as the number of gray levels, and intensity rescaling using the 64 gray levels between the absolute minimum and maximum values in the VOIs.

The CV is automatically calculated for each tumor with the mean value (μ) and standard deviation (SD, σ) obtained from a manually defined VOI. The formula is as follows: CV = σ/μ. Additionally, we harmonized the scanner effects using the ComBat method to remove data variability among MRI models [20].

The CV for each patient was calculated as the mean value for up to five tumors. If there were six or more intrahepatic lesions per patient, a maximum of five lesions were included. Patients were classified into high-CV and low-CV groups, with the median of all patients’ CV as the cutoff. We compared the rate of tumor progression between the high-CV and low-CV groups based on RECIST version 1.1.

### 2.4. Imaging Analysis: RER (Tumor/Liver Ratio)

The RER (tumor/liver ratio (T/L)) was calculated based on the SI in the pre-contrast and post-contrast MR images. Tumor lesions and the surrounding background liver were measured by defining the regions of interest manually according to previously reported methods [21]. The RER (T/L) was calculated as follows: (nodule SI/parenchyma SI on HBP)/(nodule SI/parenchyma SI in the pre-contract phase). Hyperintensity was defined as an RER ≥ 0.9 [22]. Image analysis was performed by two abdominal radiologists (N.M. and T.T., with 28 and 9 years of experience, respectively), in consensus. For each patient, the mean RER was calculated for up to five tumors.

### 2.5. Evaluation of Treatment Response

Treatment response was assessed using Gd-EOB-DTPA-MRI or CECT within 6–10 weeks after drug initiation. To determine the treatment efficacy, Gd-EOB-DTPA-MRI was performed with the same contrast as the pretreatment MRI. CECT was performed using 80–150 mL of non-ionic iodinated contrast medium (300–370 mg I/mL) administered intravenously at 2.8–4.1 mL/s with an automated injector system followed by the acquisition of images in 2–4 phases. The arterial, portal, and delayed phases were scanned at 40, 70–80, and 120–180 s after the injection of the contrast medium, respectively.

The tumor response was evaluated based on the revised RECIST guidelines (version 1.1) by radiologists with 12 and 9 years of experience specializing in abdominal imaging (K.M. and T.T., respectively) in consensus [23,24]. The patients were classified into progression (PD) and non-progression (CR, PR, and SD) groups.

### 2.6. Statistical Analysis

All statistical analyses were performed using SPSS version 26 (IBM Corp., Armonk, NY, USA). Univariate and multivariate analyses were performed using logistic regression models to analyze the prognostic factors for tumor progression in patients with u-HCC who received AB therapy. Clinical factors and rates of tumor progression were compared between the high-CV and low-CV groups using Fisher’s exact test. PFS was assessed using the Kaplan–Meier method, and the log-rank test was used to compare the two groups. Statistical significance was set at *p* < 0.05.

## 3. Results

### 3.1. Patient Characteristics

A total of 46 patients with 142 nodules were enrolled in this study. There were 35 men (76.1%) and 11 women (23.9%) with a median age of 72 years (range, 45–87 years). The background liver disease was viral hepatitis B or C in 18 patients (39.1%) and non-alcoholic steatohepatitis (NASH) in seven patients (15.2%). In the BCLC staging system, 31 patients (67.4%) were in stage B and 15 (32.6%) were in stage C. The median tumor diameter was 33.5 mm (range, 12–140 mm). Baseline patient characteristics are shown in Table 1. Gd-EOB-DTPA-MRI was performed before AB therapy (median time before AB therapy, 19 d; range, 1–41 d).

### 3.2. Treatment Response to AB Therapy

Treatment response was evaluated using Gd-EOB-DTPA-MRI in 24 patients (52%) and CECT in 22 patients (48%), with the following results: objective response rate, 6.5% (three patients); disease control rate, 71.7% (33 patients); PD, 13 patients (28.3%); SD, 30 patients (65.2%); PR, three patients (6.5%). None of the patients showed a complete response.

### 3.3. Prognostic Outcome of AB Therapy

Kaplan–Meier analysis of PFS and OS in patients with u-HCC treated with AB therapy is shown in Figure 2a and Appendix A. The median PFS and OS were 4.3 and 14.9 months, respectively. Treatments after AB therapy included the best supportive care (17 patients), lenvatinib (9 patients), AB therapy continuation (8 patients), hepatic arterial infusion chemotherapy (HAIC) (6 patients), TACE (6 patients), ramucirumab (4 patients), radiation therapy for bone metastasis (2 patients), and cabozantinib (1 patient).

The median PFS durations were 6.3 and 2.2 months (*p* = 0.002) in the non-progression and progression groups, respectively (Figure 2b). Patient characteristics in both groups are shown in Table 2. There were no significant differences in patient backgrounds, such as age, sex, background liver, or hepatic reserve capacity, between the groups.

### 3.4. Multivariate Analyses of Prognostic Factors

As shown in Table 3, CV was an independent prognostic factor for tumor progression in response to AB therapy in patients with u-HCC. Among 13 prognostic factors, the univariate analysis revealed two independent factors to be correlated with tumor progression: high CV (*p* = 0.029) and a BCLC up to-11 out (*p* = 0.081). Multivariate analysis showed that a high CV was significantly associated with tumor progression (*p* = 0.043). RER (T/L) was not a significant predictive factor in the univariate analysis.

### 3.5. Comparison Clinical Outcome Based on CV

The minimum, maximum, and median CV values for all patients were 0.13, 0.44, and 0.22, respectively. The median value of 0.22 was used as the cutoff for classification as high or low CV.

The rate of tumor progression was compared between the high- and low-CV groups; 10 patients (43.5%) in the high-CV group had tumor progression compared to three patients (13.0%) in the low-CV group. High CV (≥0.22) was significantly more frequent in the progression group (*p* = 0.047) (Table 4). Representative cases of tumors with high CV are shown in Figure 3.

PFS was compared between the high- and low-CV groups. The median PFS durations were 6.7 and 3.5 months (*p* = 0.071) in the low- and high-CV groups, respectively. Patients with a high CV tended to have a shorter PFS than those with a low CV (Figure 2c).

## 4. Discussion

This study investigated whether CV is an independent predictive factor of tumor progression for AB therapy with u-HCC. The multivariate analysis revealed that a high CV was an independent prognostic factor for AB therapy in u-HCC. RER, AFP, modified ALBI, and extrahepatic metastasis, which have been previously reported to be significantly correlated with the treatment response to AB therapy, did not show any significant differences [7,25,26,27]. Imaging biomarkers may be more reliable than biochemical indicators for predicting treatment efficacy.

Minamiguchi et al. reported that a CV ≥ 0.16 lead to worse prognosis following TACE in intermediate-stage HCC [14]. Since our study was conducted on u-HCC, which is a different subject from the previous study, a new cutoff value needs to be established. CV ≥ 0.22, the median value for all patients, was applied in this study as a cutoff, as in previous studies [28,29,30,31].

HCC typically shows homogeneous low SI on HBP images, but sometimes shows heterogeneous or high SI on HBP images. An anatomical structure or biochemical processes affecting signal intensity are still not fully understood. Previous reports showed organic anion-transporting polypeptides-8 (OATP8) expression in HCCs with heterogeneous hyperintensity was significantly higher than in HCCs with homogeneous hypo-intensity [32]. This suggests that the Wnt/β-catenin signal associated with the OATP expression may be a process involved in the heterogeneous hyperintensity. Some studies have indicated that HCCs displaying heterogeneous SI on the HBP exhibit greater resistance to immunotherapy than other HCCs [32,33,34,35]. Although the reason for the greater malignant potential of HCCs with signal heterogeneity remains unclear, Fujita et al. proposed that the SI on HBP images finally changes to heterogeneous hyperintensity with increasing degree of malignancy [32].

In our study, we demonstrated the usefulness of CV, which reflects the signal heterogeneity of HBP, in predicting the initial treatment response among patients with u-HCC undergoing AB therapy. It was recently reported that HCCs showing heterogeneous SI have a shorter PFS after AB therapy than HCCs showing homogeneous SI, as heterogeneous HCC may reflect the degree of differentiation and non-uniformity of molecular biological characteristics [7]. In our study, CV, which exhibits heterogeneous SI, may be a poor prognostic factor for AB therapy in u-HCC. However, the reason that HCCs in the high-CV group showed a poorer response to AB therapy requires further molecular pathological investigation.

There are three calculation methods for RER: RER (T/L), RER (SI), and RER (T1). The RER (T/L) was calculated as (tumor SI in HBP/liver SI in HBP)/(tumor SI in pre-contrast/liver SI in pre-contrast). The RER (SI) was calculated as (tumor SI in HBP—tumor SI in pre-contrast)/(tumor SI in pre-contrast). The RER (T1) is calculated as (1/tumor T1 value in HBP-1/tumor T1 value in pre-contrast)/(1/tumor T1 value in pre-contrast). Among these methods, RER (T1) is the most accurate; however, we could not calculate the RER (T1) at our institution but Kitao et al. calculated the T1 value using the double-flip angle method of MRI [36]. Nevertheless, the RER (T/L) has been reported as a useful indicator for predicting the efficacy of immunotherapy. HCC with a high RER (T/L) in the HBP has been reported to be resistant to immunotherapy [5,6]. Ueno et al. reported that β-catenin–activated HCC is detectable by RER (T/L), with a sensitivity of 78.9% and a specificity of 81.7% [22,37]. In our study, RER (T/L) was not a significant factor in the initial treatment effect of AB therapy (*p* = 0.667); however, the percentage of HCC with high RER was only 10.8%.

In immunotherapy, two events make it difficult to determine the therapeutic efficacy: hyper-progression and pseudo-progression. Our results showed that the initial treatment response influenced PFS after AB therapy. The iRECIST recommends re-evaluation after immune-unconfirmed PD of the initial treatment response to prevent the underestimation of the treatment response to immunotherapy due to pseudo-progression [38]. However, in a previous report, there was no difference in the percentage of the disease control rate between the initial and best treatment response by RECIST version 1.1 in u-HCC treated by AB therapy [25]. Furthermore, it has been reported that more than 90% of u-HCC treated by immunotherapy did not show pseudo-progression [39].

Recent studies have reported the clinical benefits of lenvatinib compared to AB therapy. Casadei-Gardini et al. reported that OS was prolonged by lenvatinib compared to AB therapy in patients with NASH/non-alcoholic fatty liver disease [40]. Rimini et al. reported that lenvatinib as a first-line treatment resulted in a significantly longer OS than AB therapy in patients with Child–Pugh class B disease [41]. Three-dimensional CV may also be applied to determine the usefulness of molecular-targeted agents, as heterogeneous SI on HBP images does not affect PFS in patients with u-HCC treated with lenvatinib [7].

Our study had several limitations. First, because of the focus on tumor signal heterogeneity in the HBP, the process of calculating CV had to be limited to intrahepatic lesions. Second, the diagnosis of u-HCC at the start of treatment was made radiologically, with the absence of histopathological diagnosis. Third, the VOIs were established through a collaborative agreement between the two readers. A comprehensive investigation is necessary to evaluate the interobserver concordance in VOI placement, thereby addressing the issue of reader variability. Fourth, parameters were assessed in images obtained with scanners of different magnetic field strengths (1.5 and 3.0 T), but we harmonized with the ComBat method. Fifth, we did not obtain the histopathological and immunohistochemistry findings (i.e., β-catenin and hepatocyte nuclear factor 4α). Finally, although this study was a multi-center retrospective analysis, the relatively small sample size highlights the need for larger prospective validation studies to corroborate our findings.

## 5. Conclusions

Quantitative analysis of signal heterogeneity in the HBP of Gd-EOB-DTPA MRI is useful for predicting tumor progression in AB therapy. The CV of the HBP could be an imaging biomarker that can contribute to the individualization of therapeutic strategies in u-HCC.

## Figures and Tables

**Figure 1 cancers-16-02275-f001:**
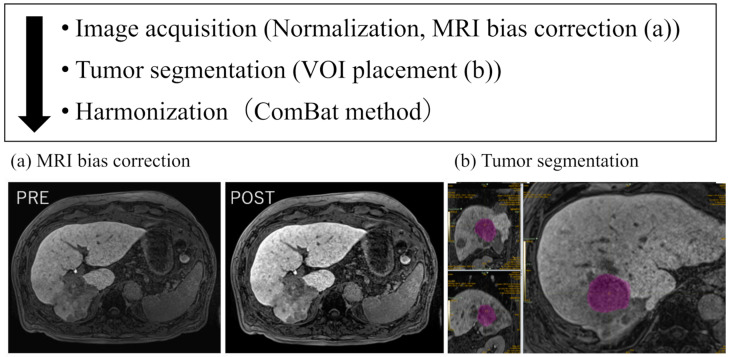
Flowchart of radiomic analysis. Radiomic analysis workflow to extract CV in u-HCC. The N4 bias field correction algorithm was applied to correct the inhomogeneity of the images using 3D Slicer software (**a**). We drew the volumes of interest to include the whole tumor volume on multiple slices using LIFEx software (v. 7.3.0) (**b**). We focused on the “Intensity-Based-Coefficient of Variation” as CV of the whole tumor. We harmonized the scanner effects using the ComBat method to remove data variability between models. CV, coefficient of variation; u-HCC, unresectable hepatocellular carcinoma.

**Figure 2 cancers-16-02275-f002:**
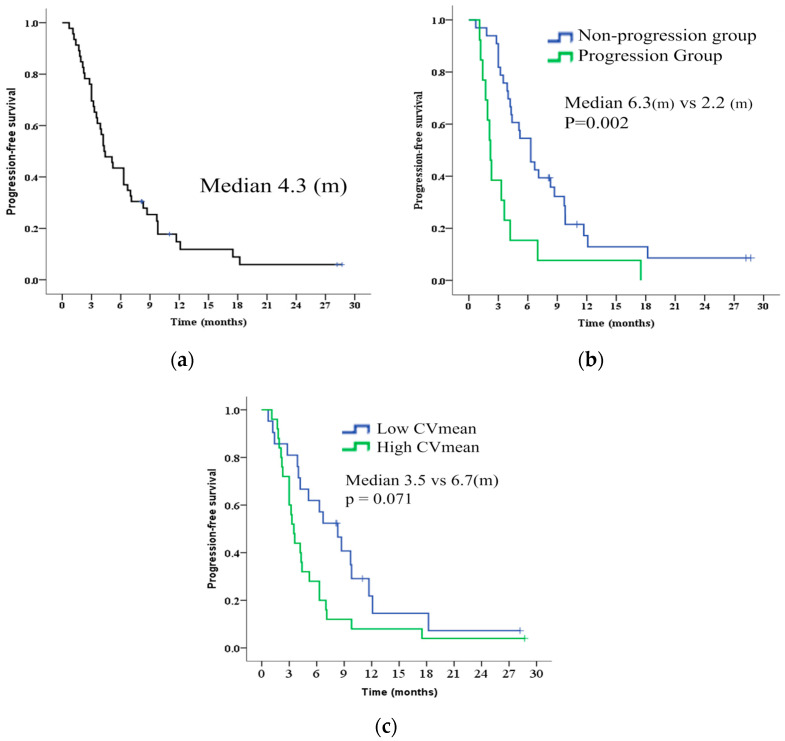
Kaplan–Meier analysis of PFS in all patients, comparison between progression and non-progression groups for AB therapy, and comparison between high and low CV. (**a**) The median PFS was 4.3 months in all patients. (**b**) The median PFS was 2.2 and 6.3 months (*p* = 0.002) in progression and non-progression groups, respectively. (**c**) The median PFS was 6.7 and 3.5 months (*p* = 0.071) in the low and high CV groups. PFS, progression-free survival; AB, atezolizumab and bevacizumab; u-HCC, unresectable hepatocellular carcinoma; CV, coefficient of variation.

**Figure 3 cancers-16-02275-f003:**
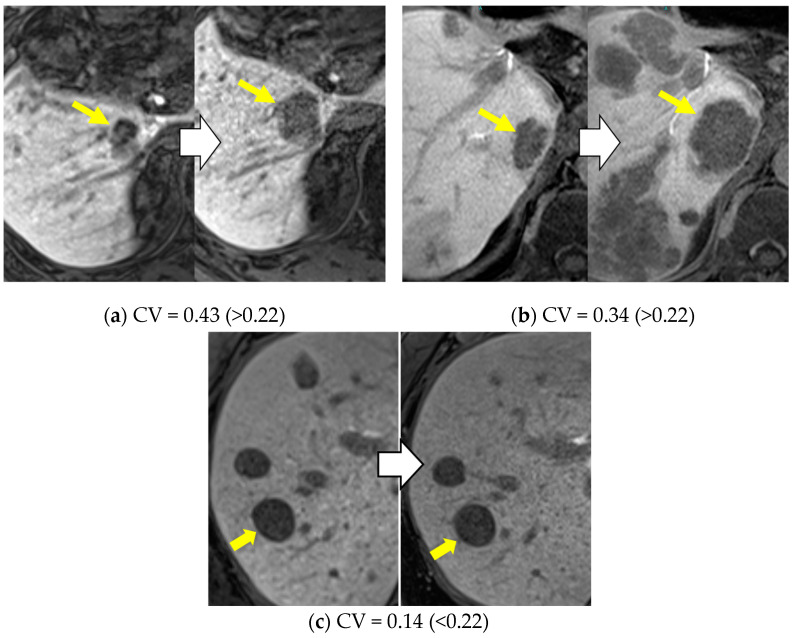
Representative cases of u-HCC classified as high and low CV. Two patients with u-HCC were classified as having high CV. The hepatobiliary phase of Gd-EOB-DTPA-MRI shows visually heterogeneous tumor (**a**) and visually homogenous but quantitatively heterogeneous tumor (**b**). These nodules with high CV were enlarged and determined to be in the progression groups. On the other hand, the visually homogenous and low-CV tumor had no progression for AB therapy (**c**). u-HCC, unresectable hepatocellular carcinoma; CV, coefficient of variation; Gd-EOB-DTPA-MRI, gadolinium ethoxybenzyl-diethylenetriaminepentaacetic enhanced magnetic resonance imaging; AB, atezolizumab/bevacizumab.

**Table 1 cancers-16-02275-t001:** Characteristics of 46 patients with u-HCC.

Characteristic		All Patients (*n* = 46)
Age (year), *n* (%)	<70	16 (34.8%)
	≥70	30 (65.2%)
Sex, *n* (%)	Male	35 (76.1%)
	Female	11 (23.9%)
Etiology of liver disease, *n* (%)	Viral hepatitis	18 (39.1%)
	NASH	7 (15.2%)
	Alcohol	12 (26.1%)
	None	10 (21.7%)
BCLC guideline, *n* (%)	B (7 in)	2 (4.3%)
	B (7–11)	11 (23.9%)
	B (11 out)	18 (39.1%)
	C	15 (32.6%)
Previous treatment history for HCC, *n* (%)	Partial hepatectomy	12 (26.1%)
	RFA (radiofrequency ablation)	9 (19.6%)
	Lip-TACE	24 (52.2%)
	Bland-TAE	6 (13.0%)
	DEB-TACE	1 (2.2%)
	Lenvatinib	20 (43.5%)
	Ramucirumab	3 (6.5%)
	Sorafenib	3 (6.5%)
	Cabozantinib	1 (2.2%)
	HAIC (hepatic arterial infusion chemotherapy)	4 (8.7%)
	Radiation therapy (RT)	1 (2.2%)
	PEIT (percutaneous ethanol injection therapy)	1 (2.2%)
	None	8 (17.4%)
mALBI grade, *n* (%)	1	17 (37.0%)
	2a	12 (26.1%)
	2b	15 (32.6%)
	3	2 (4.3%)
Child-Pugh, *n* (%)	A	41 (89.1%)
	B	5 (10.9%)
PT (%), *n* (%)	<70	3 (6.5%)
	≥70	43 (93.5%)
Total bilirubin (mg/dL), *n* (%)	<2	45 (97.8%)
	≥2	1 (2.2%)
Albumin (g/dL), *n* (%)	<3.5	12 (26.1%)
	≥3.5	34 (73.9%)
AFP (ng/mL), *n* (%)	<400	31 (67.4%)
	≥400	15 (32.6%)

HAIC, hepatic arterial infusion chemotherapy; RT, radiation therapy; PEIT, percutaneous ethanol injection therapy.

**Table 2 cancers-16-02275-t002:** Comparison of the patient characteristics between progression and non-progression groups for AB therapy.

Characteristic		Progression Group (*n* = 13)	Non-Progression Group (*n* = 33)	*p* Value
Age, *n* (%)	<70	4 (30.8%)	12 (36.4%)	1.000
	≥70	9 (69.2%)	21 (63.6%)	
Sex, *n* (%)	Male	12 (92.3%)	23 (69.7%)	0.141
	Female	1 (7.7%)	10 (30.3%)	
Etiology of liver disease, *n* (%)	Viral hepatitis	6 (46.2%)	12 (36.4%)	0.738
	NASH	1 (7.7%)	6 (18.2%)	0.654
	Alcohol	4 (30.8%)	8 (24.2%)	0.717
	NonBnonC	3 (23.1%)	7 (21.2%)	1.000
BCLC guidelines, *n* (%)	B (7 in)	0 (0.0%)	2 (6.1%)	0.271
	B (7–11)	1 (7.7%)	10 (30.3%)	
	B (11 out)	6 (46.2%)	12 (36.4%)	
	C	6 (46.2%)	9 (27.3%)	
mALBI grade, *n* (%)	1	7 (53.8%)	10 (30.3%)	0.218
	2a	1 (7.7%)	11 (33.3%)	
	2b	5 (38.5%)	10 (30.3%)	
	3	0 (0.0%)	2 (6.1%)	
Child–Pugh, *n* (%)	A (5)	7 (53.8%)	21 (63.6%)	0.157
	A (6)	6 (46.2%)	7 (21.2%)	
	B	0 (0.0%)	5 (15.2%)	

BCLC guidelines, Barcelona Clinic Liver Cancer guidelines; NASH, non-alcoholic steatohepatitis.

**Table 3 cancers-16-02275-t003:** Comparison of patient characteristics between progression and non-progression groups for AB therapy.

	Univariate Analysis				Multivariate Analysis			
	*p*-Value	OR	95% CI		*p*-Value	OR	95% CI	
Age ≥ 70	0.720	1.286	0.325	5.084				
Sex	0.136	0.192	0.022	1.68				
Liver disease (NASH)	0.387	0.375	0.041	3.465				
Child–Pugh score 5	0.541	1.500	0.408	5.508				
Modified ALBI grade 1	0.143	0.373	0.100	1.394				
BCLC B vs. C	0.224	2.286	0.603	8.665				
BCLC B (UT-7 in and 7–11) vs. B (UT-11 out) and C	0.081	6.857	0.791	59.446	0.105	6.258	0.682	57.424
AFP ≥ 400 ng/mL	0.391	0.525	0.120	2.289				
CV ≥ 0.22	0.029	5.128	1.183	22.238	0.043	4.786	1.053	21.746
RER ≥ 0.90	0.667	0.604	0.061	5.98				
Largest tumor diameter ≥ 40 mm	0.540	0.664	0.179	2.460				
Extrahepatic lesion	0.813	1.185	0.291	4.830				
Previous TACE (lipiodol, bland, beads)	0.743	0.800	0.211	3.029				

NASH, non-alcoholic steatohepatitis; BCLC, Barcelona Clinic Liver Cancer guidelines; AFP, alfa fetoprotein; CV, coefficient of variation; RER, relative enhancement ratio; TACE, trans-arterial chemoembolization.

**Table 4 cancers-16-02275-t004:** Comparison of CV between progression and non-progression groups.

	High CV	Low CV	Total
	(*n* = 23)	(*n* = 23)	
Non-progression group (SD, PR, CR)	13 (56.5%)	20 (87.0%)	33
Progression group (PD)	10 (43.5%)	3 (13.0%)	13

CV, coefficient of variation; SD, stable disease; PR, partial response; CR, complete response; PD, progressive disease *p* = 0.047.

## Data Availability

The datasets generated and/or analyzed during the current study are available from the corresponding author upon reasonable request.

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
