# Peer review of "Prediction of Efficacy for Atezolizumab/Bevacizumab in Unresectable Hepatocellular Carcinoma with Hepatobiliary-Phase Gadolinium Ethoxybenzyl-Diethylenetriaminepentaacetic Acid MRI"

_cancers, 2024, doi:10.3390/cancers16122275_

Round 1

Reviewer 1 Report

Comments and Suggestions for Authors

The article entitled ”Prediction of efficacy for Atezolizumab/Bevacizumab in unresectable hepatocellular carcinoma with Hepatobiliary-phase gadolinium ethoxybenzyl-diethylenetriaminepentaacetic acid MRI” by Kunichika et al. aims to investigate in a retrospective cohort the predictive potential of the coefficient of variation (CV) of Gd-EOB-DTPA signal intensity in HCC tumor progression after AB immuno-/-antiangiogenic therapy.

The manuscript is well-written. Methodology appears solid and plausible conclusions were drawn. However, the article would benefit from some representative images of low vs. high CV group to visualize the contrast agent signal heterogeneity. Moreover, lack of histological diagnosis of a pathologist harbors some limitation. Different histological subtypes could have some impact on signal intensity and heterogeneity.

The following points should be addressed:

1.   Is it known yet which anatomical structure or biochemical processes influence the heterogeneous signal intensity of the contrast agent Gd-EOB-DTPA in the liver? Please add some details to the discussion, if available.

2.   Page 2, Graphical Abstract: The Kaplan-Meier curve shows no significant (p = 0.071) difference between low and high CV groups, but a tendency. Have the authors tried another cutoff than all patients’ median CV value (e.g. 5 % higher or lower than the median)?

3.   Page 3, line 107: A confirmation of HCC diagnosis by a pathologist is highly recommended. Ideally, CV only of tumors of the same histological growth pattern (trabecular, pseudoglandular and solid) should be compared since different histological subtype might lead to varying signal intensity. Please add the lack of histological diagnosis to the limitations paragraph (page 10, line 340-350).

4.   Page 3, line 120: How and how often where the patients treated with bevacizumab? Also IV and once every three weeks? Please add this information.

5.   Page 4, Figure 1: Images (a) and (b) were not described in the caption. Please change the title of Figure 1 and add some information for (a) and (b). What is shown here? What is represented by the pink area? Explain.

6.   Page 4, line 160 and 173: “up to five tumors”. Do the authors mean ”five tumor images” or did each patient harbor more than five individual tumors? Please clarify in the text.

7.   Page 6, line 224-228: Have the authors looked for significant differences in PFS based on patients’ treatment history (if statistically possible)?

8.   Page 9, Figure 3: The authors show only a high CV case. Please, show also an image a low CV case of the non-progression group for direct comparison.

9.   Supplementary Table 1: “1” is missing. Why were the institutions anonymized (1-6)?

Author Response

Thank you very much for taking the time to review this manuscript. Please find the detailed responses below and the corresponding revisions in track changes in the re-submitted files.

Reviewer 2 Report

Comments and Suggestions for Authors

This Japanese study describes the utility of coefficient of variation in gadolinium MRI in HCC patients receiving immune therapy

My comments:

The study is of some interest

The sample size is rather small. How can you defend the number of variables used for such a small study?

Was the CV >/= 0.22 predefined? If not then you need a validation set of patients

Are you planning prospective validation study

Author Response

(The authors gave the same response as above.)

Reviewer 3 Report

Comments and Suggestions for Authors

Comments:

The author of this study investigated the value of coefficient of variation (CV) in the hepatobiliary-phase (HBP) of Gd-EOB-DTPA-MRI could be an independent predictive factor for tumor progression. Their results indicated that quantitative analysis using CV in the HBP may be useful for predicting tumor progression for Atezolizumab/bevacizumab therapy. The subject of this manuscript is very interesting, but there are a few of minor defects need to be modified.

1. When the term first appears, the full name and abbreviation need to be provided, and then the abbreviation should be used. Please check:Simple Summary section:Atezolizumab/bevacizumab (AB) therapy is currently one of the first-line drugs for patients with unresectable hepatocellular carcinoma (u-HCC). Predicting early progressive disease before AB therapy may be beneficial for selecting alternative treatments for patients with unresectable hepatocellular carcinoma. Coefficient of variation, as one of the quantitative indicators of signal heterogeneity, in the hepatobiliary phase of Gd-EOB-DTPA-MRI was an independent predictive factor for tumor progression (p = 0.043)......Line 2-3:unresectable hepatocellular carcinoma should be changed to:u-HCC; Coefficient of variation should be change to: Coefficient of variation (CV).

2. Abstract section: HBP should be changed to: HBP of Gd-EOB-DTPA-MRI. 

3. Imaging analysis CV section: The method of obtaining CV should be described in more detail.

4. The author should explain why RECIST was used instead of modified RECIST to evaluate treatment response.

Author Response

(The authors gave the same response as above.)

Reviewer 4 Report

Comments and Suggestions for Authors

Kunichika et al present a multicenter cohort study analyzing the progression-free survival of 46 patients with unresectable HCC receiving atezo/beva. In a first part, they analyze risk factors associated with progressive disease, identifying a "high" coefficient of variation (CV) in the hepatobiliary phase of primovist MRI as an independent predictor through multivariate analysis. In a second phase, they analyze the probability of progression over time according to a baseline "high vs low" CV, showing a trend towards higher probabilities of developing progression over time.

This is a well written manuscript. Most of the authors already participated in a paper published last year in Cancers. I have only minor comments:

- The authors refer the association of relative enhancement ratio (RER) as a radiogenomic marker of B-catenin activation, which could be associated with some form of resistance to immunotherapy. However, even though they analyzed RER, it was not found to be significant on multivariate analysis. CV, on the other hand, was a significant factor. Several lines are dedicated in the discussion to RER but given the fact that it is not associated and it doesn´t explain the mechanism by which a high CV is associated with worst prognosis, there is no point in keeping them.

- The manuscript does not state clearly, how CV become the focus of this study. In their previous manuscript, Minamiguchi et al analyzed the impact of "high" CV on the mean survival time after TACE for intermediate HCC. They used this marker, based on a previous study by a Korean group (Lee et al) who analyzed twelve different radiological markers, identifying signal heterogeneity (qualitatively) during HBP on primovist MRI as a prognostic marker. There is currently no mechanism by which a high CV explains the biological behavior of HCC, therefore, the authors should refrain from associating it with resistance to immunotherapy. It may just be a poor prognostic marker overall.

- In their previous study, Minamiguchi et al defined the method by which they established a CV cutoff of >0.16 to define "high CV". In the current manuscript, the methods section does not explain why the authors established a CV cutoff of >0.22. This methodology should be explained in the current manuscript.

- Figure 4 shows the development of progressive disease over time according to the CV count. It may as well be included as Figure 2c to facilitate the reading of the manuscript.

- The supplemental figure shows the survival of all patients in this study, showing a mean survival of 14.9 months. Given the fact that the previous study by the same group of authors demonstrated a significantly reduced mean survival after TACE in patients with a high CV, patient survival should be included in the result section and a supplementary figure added showing the likelihood of death overtime according to high vs low VC.  

- In the discussion section, it is stated that "The reason the HCCs in the high-CV group showed a poorer response to AB therapy requires further pathological investigation". This is a correct statement, and it should be expanded to recognize that a high CV may be a nonspecific marker of poor prognosis.

I congratulate the authors for this manuscript.

Author Response

(The authors gave the same response as above.)
